# A Real-time Contribution Measurement Method for Participants in Federated Learning

## Abstract

Federated learning is a framework for protecting distributed data privacy and has participated in commercial activities. However, there is a lack of a sufficiently reasonable contribution measurement mechanism to distribute the reward for each agent. In the commercial union, if there is no mechanism like this, every agent will get the same reward. This is unfair to agents that provide better data, so such a mechanism is needed. To address this issue, this work proposes a real-time contribution measurement method. Firstly, the method defines the impact of each agent. Furthermore, we comprehensively consider the current round and the previous round to obtain the contribution rate of each agent. To verify effectiveness of the proposed method, the work conducts pseudo-distributed training and an experiment on the Penn Treebank dataset. Comparing the Shapley Value in game theory, the comparative experiment result shows that the proposed method is more sensitive to both data quantity and data quality under the premise of maintaining real-time.

## 1 Introduction

There are lots of data generate, collect, and access every day by smart terminals. But cause of the privacy of these data, it is usually difficult to use them. Such as the language model to predict the next word or even entire reply(Ion et al., 2016).

The emergence of federated learning breaks this data barrier. It can use agent computing power to conduct model training while maintaining data localization and privacy protection, and obtain an excellent global model.

But in a commercial federation, each agent should get corresponding rewards based on its contribution to the model, not the same rewards. There are many methods for contribution measurement. Such Wang et al. (2019) measured the contribution of each group features in vertical federated learning, and Zhan et al. (2020) proposed an incentive mechanism to make each agent willing to contribute better data. But most of them need to consume large computing resources and many methods are calculated offline.

In order to address this problem, this paper proposes a method for obtaining the contribution of each agent in real time with a small amount of calculation in horizontal federated learning.

Our contributions in this paper are as follows:

- We propose a method to measure agents' contributions and compare this method with Shapley Value.

- The method we propose is sensitive to data volume and data quality, and can be used for mutual comparison between agents.

- In the training process, the contribution to each agent can be obtained in real time, with low computational complexity.

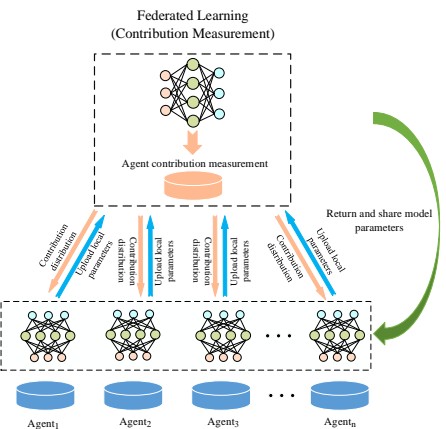

Figure 1: Experimental results with randomize the word sequence.

## 2 RELATED WORK

In this section, we will introduce the current research situation and application of federated learning, as well as the existing problems of reward distribution in the commercialization of federated learning framework, and introduce the necessity and importance of our work.

### 2.1 FEDERATED LEARNING

Distribute the training data on each mobile device to maintain the localization of the data, instead of transmitting the data to the central server, updating the model locally, and uploading the update results to the server. While maintaining data localization and privacy, it can aggregate the data of each agent. The central server collects agent data and uses FedSGD, FedAVG(Brendan McMahan et al., 2016) and other algorithms to maintain the central model in combination with the different optimizer(Felbab et al., 2019), and sends the updated model to each agent. During the transmission process, methods such as homomorphic encryption are used to protect the security of data transmission and maintain the continuous iterative update of the model. This method is federated learning.

At present, for different datasets, federated learning framework can be classified into horizontal federated learning, vertical federated learning, and federated transfer learning(Yang et al., 2019). Horizontal federated learning is suitable for situations where the data provided by the agents has more of the same characteristics. In (McMahan et al., 2017), Google proposed a solution to update the horizontal federated learning model in Android phones. vertical federated learning is suitable for situations where there is less feature overlap but more user id overlap. Hardy et al. (2017) proposed a vertical federated learning method for training a logistic model. Chen et al. (2020) proposed the FedHealth method, which uses federated learning to aggregate data and uses transfer learning to obtain a personalized model.

Under the framework of federated learning, there are many different directions of research. Aono et al. (2017) use the data transmitted by the homomorphic encryption agents and the central server for model training, which further strengthens the privacy of the agents' data. In (Lu et al., 2019; Kim et al., 2019), data verification is carried out in conjunction with the blockchain to prevent the gradient information from being maliciously tampered with. Konečnỳ et al. (2016) research on reducing the consumption of communication resources in federated learning.

### 2.2 APPLICATION AND COMMERCIALIZATION OF FEDERATED LEARNING

Since federated learning was proposed, federated learning has been successfully applied to more and more scenarios. When considering data privacy issues, many companies will choose to use federated learning to protect data privacy to achieve cooperation. Such WeBank has successfully used federated learning in bank federations for credit evaluation and other financial aspect. In (Ren et al., 2019), federated learning is applied to dynamic IoT system.Lu et al. (2019); Kim et al. (2019)

use blockchain for data verification as an alternative to the commercialization of federated learning. Bao et al. (2019) achieved model commercialization by providing a healthy marketplace for collaborative-training models.

In the commercialization process of federated learning, the method of using this decentralized training model provides privacy, security, supervision and economic benefits, but also brings many fairness and systemic challenges.

### 2.3 Contribution Measurement and Incentive Mechanism in Federated Learning

Wang et al. (2019) proposed grouped instance influence to measure contribution of agents in horizontal federated learning and grouped feature importance in vertical federated learning which equitably and accurately realized the measurement of contributions of all agents. But this method cannot take into account the amount of data.

Geyer et al. (2017) proposed a federated learning optimization algorithm based on differential privacy protection of the agents that can hide the contribution of the agents. In order to balance the contribution of the soft training model and ensure the collaborative convergence, Xu et al. (2019) proposed the corresponding parameter aggregation scheme. Kang et al. (2019) used the weighted subjective logic model to design a reputation-based worker selection plan to achieve reliable federated learning. In (Zhan et al., 2020), designed an incentive mechanism based on deep reinforcement learning to determine the optimal pricing strategy for parameter servers and the optimal training strategy for agents.

Our work focus on real-time contribution measurement mechanism for each agent in vertical federated learning. Current research has problems such as poor real-time performance and high resource consumption by contribution measurement methods. The method proposed in this paper can obtain real-time contribution measurement to each agent in the process of federated learning. At the same time, it verifies that the method proposed in this paper has high sensitivity on data quantity and data quality.

## 3 Methodology

In order to solve the above issues, we propose a corresponding method which does not need to obtain agents data and data scale, but can measure agents contribution. This method of measuring agents contributions is suitable for evaluating agents in real time.

### 3.1 Federated Learning

In this section, we will introduce the basic framework of federated learning and how parameters are updated.

Federated learning is a distributed learning method in which the server maintains an central model and distributes it to individual agents. To solve privacy problems, the server does not need to obtain the data of the agents, so the computing power of the agents is used to learn at the agents local environment.

The server will set up a fraction $C$, to select the agents proportionally for the server's central model update. Then the selected agents' updated model parameters are uploaded to the server for parameter updating of the server model. It is then distributed to the each agent to improve the model of the agents.

In this way, it continue to improve the central model of the server and the local model of the agents. Under the premise of ensuring the correctness and privacy of the agents, using this method can make full use of the computing power of the agents and a large amount of private data for learning, and at the same time maintain an excellent global model.

## 3.2 ATTENTION AGGREGATION

In this section, we will introduce a FedAtt algorithm proposed by Shaoxiong Ji, Shirui Pan, Guodong Long et al.Ji et al. (2019), which is outperformed to the common FedAvg and FedSGD algorithm. The proposed attention mechanism in Federated Learning is used after the agents transmitted the parameters to the server, the server uses Formula equation 1 to calculate the value of attention $\alpha$ of each layer parameter that should be allocated to the agent, and multiplies each layer parameter by the corresponding attention to update the central model. The server integrates the update parameters passed by the agents according to theirs own parameter information, and finally completes an update of the central model through the communication between the server and the agents.

$$\alpha_k^l = \text{softmax}(s_k^l) = \frac{e^{s_k^l}}{\sum_i e^{s_i^l}} \tag{1}$$

The $s_k^l$ is the norm difference from the central model. We use $w^l$ to represent layer $l$ parameters of the server and $w_k^l$ for $l$th layer parameters of the agent model $k$. So the definition of $s_k^l$ is as follows

$$s_k^l = ||w^l - w_k^l||_p \tag{2}$$

So for $m$ agents that are selected to update the central model, the method of updating becomes

$$w_{t+1} \leftarrow w_t - \epsilon \sum_{k=1}^m \bigtriangledown(w_t^k) = w_t - \epsilon \sum_{k=1}^m \alpha_k(w_t - w_t^k) \tag{3}$$

To protect the agents' data privacy, you can add the randomized mechanism before the agent passes parameters to the server. Randomly generate a random vector that obeys the standard distribution $\mathcal{N}(0, \sigma^2)$, multiply the corresponding weight $\beta$, the results of the final update parameters as shown in Formula equation 4

$$w_{t+1} \leftarrow w_t - \epsilon \sum_{k=1}^m \alpha_k(w_t - w_t^k + \beta \mathcal{N}(0, \sigma^2)) \tag{4}$$

The implementation process of the whole algorithm is shown as Algorithm 1

---

**Algorithm 1** Attentive Federated Optimization

1: $l$ is the ordinal of neural network layers; $\epsilon$ is the stepsize of server optimization
2: **Input:** server parameters $w_t$ at $t$, agents parameters $w_{t+1}^1, w_{t+1}^2, \cdots, w_{t+1}^m$ at $t+1$
3: **Output:** server parameters $w_{t+1}$ at $t+1$
4: Initialize attention $\alpha = \{\alpha_1, \alpha_2, \cdots, \alpha_m\}$
5: **for** each layer $l$ in model **do**
6:     **for** each agents $k \in S_t$ from 1 to $m$ **do**
7:         $s_k^l = ||w^l - w_k^l||_p$
8:     **end for**
9:     $\alpha_k^l = softmax(s_k^l) = \frac{e^{s_k^l}}{\sum_i e^{s_i^l}}$
10: **end for**
11: $\alpha_k = \{\alpha_k^1, \alpha_k^2, \cdots, \alpha_k^l\}$
12: $w_{t+1} \leftarrow w_t - \epsilon \sum_{k=1}^m \alpha_k(w_t - w_t^k + \beta \mathcal{N}(0, \sigma^2))$

---

## 3.3 AGENTS CONTRIBUTIONS

In this section, the method for measuring agents contribution proposed in this paper will be introduced in detail.

Before calculating each agent contribution, we make the following assumptions: we think each terminal does not tamper with the updated gradient itself during the previous process of passing and updating parameters between the server and the agents.

The agents contribution to the server should be calculated after each layer of agents' parameter attention. When the server has calculated the attention of agent $K$, it can calculate the impact of this agent $K$ on the server model parameters in the $T$ update. We define $imp_t^k$ as follows:

$$imp_t^k = \epsilon\alpha_k(w_t - w_t^k + \beta\mathcal{N}(0, \sigma^2)) + \gamma \cdot imp_{t-1}^k \tag{5}$$

where $\gamma \in (0, 1)$ is forgetting coefficient, because large variations in the early stage of the training model, the model is not stable, and the impact of the previous endpoint should be reduced after multiple iterations. If the agent $k$ is not in the selected $M$ agents this time, we think that the impact of the agent $t$ round is $imp_t^k = imp_{t-1}^k$, that is, only the number of rounds that the agent participates in the update is calculated. Note that the server's attention to the agents are the attention of each layer, so when calculating the impact it also calculates the impact of each layer by reweighting and averaging.

We will be each agents attention MinMaxScaler normalized limited range, then Softmax, to obtain the contribution of each agent. We use $con_t^k$ to represent the contribution of agent $k$ at $t$ time. The whole method of measuring agents contribution proposed in this paper is shown in Algorithm 2

---

**Algorithm 2** Measure Agents Contributions

---

1:  **Input:** $l$ is the ordinal of neural network layers; $\epsilon$ is the stepsize of server optimization, server parameters $w_t$ at time $t+1$, agents parameters $w_{t+1}^1, w_{t+1}^2, \cdots, w_{t+1}^m$ at time $t+1$, $\gamma$ is forgetting coefficient, $S_t$ is selected agents set.
2:  **Output:** Agents contributions $con_t$ at $t$
3:  **for** each agents $k$ from 1 to $K$ **do**
4:      **if** $k \in S_t$ **then**
5:          $imp_t^k = \epsilon\alpha_k(w_t - w_t^k + \beta\mathcal{N}(0, \sigma^2)) + \gamma \cdot imp_{t-1}^k$
6:      **else**
7:          $imp_t^k = imp_{t-1}^k$
8:      **end if**
9:  **end for**
10: $con_t = \text{softmax}(\text{MinMaxScaler}(imp_t))$

---

## 4 EXPERIMENT

In this section, we will introduce the verification experiments performed for our proposed agents' contribution measurement method.

### 4.1 EXPERIMENTAL ENVIRONMENT AND CONFIGURATION

The system used in our verification experiment is Ubuntu 18.04 LTS, the backend used is the GPU version of Pytorch, with the NVIDIA GTX1660Ti GPU acceleration for model calculation. The GRU-based agents model used the language processing model of RNN. For a detailed model description, see the subsection C under this Section.

### 4.2 DATASET

We perform experimental verification on public language datasets of Penn Treebank[1] to verify the effectiveness of our proposed algorithm and its sensitivity to special variables.

### 4.3 MODEL

In natural language processing, the LSTM model is often used for processing. We used a smaller GRU-based agent language model. First take texts as input, and convert words into word vectors according to a pre-built dictionary. The converted word vector is then used as an input to the LSTM model, and the prediction result is finally output.

---

[1]Penn Treebank is available at
`https://github.com/wojzaremba/lstm/tree/master/data`

### 4.4 PARAMETERS

This section enumerates the meanings and values of the various parameters mentioned. At the same time, we believe that the data of each agent is independent and identically distributed. The parameters and their descriptions are listed in Tab. 1

Table 1: Table of parameters mentioned

| Name | Represent letter | Value |
|------|------------------|-------|
| Number of agents | $K$ | 20 |
| Server training rounds | round | 10 |
| The number of local epoch | epoch | 5 |
| The fraction of agents | $C$ | 0.3 |
| Learning rate of agents | $\eta$ | 0.01 |
| Learning rate of server | $\eta'$ | 0.001 |
| Batch size | $B$ | 128 |
| Step size | $\epsilon$ | 1.2 |
| Differential privacy | $\beta$ | 0.001 |
| Embedding dimension | none | 300 |
| Forgetting coefficient | $\gamma$ | 0.9 |

### 4.5 EXPERIMENTAL RESULTS

We use testing perplexity as an indicator of the evaluation model. In information theory, perplexity is a measurement of how well a probability distribution or probability model predicts a sample. The perplexity of a discrete probability distribution is defined as:

$$ppl(x) = 2^{H(p)} = 2^{-\sum_x p(x)log_2 p(x)} \tag{6}$$

In the above formula, $H(p)$ is the expected probability distribution. If the prediction result of our model is $m(x)$, then the perplexity of the language model is defined as:

$$ppl(x) = 2^{H(p,m)} = 2^{-\sum_x p(x)log_2 m(x)} \tag{7}$$

Obviously, $H(p) \leqslant H(p,m)$. Therefore, the smaller the perplexity, the more representative the probability distribution can be to better predict the sample distribution.

And we compared with the results of Shapley Value evaluation. Shapley Value is originated from coalitional game theory and has proven theoretical properties. It provides a way to measure the impact and contribution of various agents. The definition of Shapley Value is:

$$\varphi_i(x) = \sum_{Q \subseteq S \setminus \{i\}} \frac{|Q|! - (|S| - |Q| - 1)!}{|S|!} (\Delta_{Q \cup \{i\}}(x) - \Delta_Q(x)) \tag{8}$$

$S$ is the set of all agents, $Q \subset S = 1, 2, \cdots, n$ is a subset of the agent set $S$, $i$ is the index of the agents, $||$ represents the size of the set, $\Delta_Q(x) = imp_Q$ denotes the impact of agent set $Q$.

Because the complexity of directly calculating Shapley value is too high, we use the following estimation method to get the $\varphi_i(x)$ of each Agents

$$\varphi_i(x) = \frac{1}{M} \sum_{m=1}^{M} (\Delta_{Q^m \cup \{i\}}(x) - \Delta_{Q^m}(x)) \tag{9}$$

where $M$ is the number of iterations. $\Delta_{Q^m}(x)$ denotes the impact of random set $Q^m$. Finally, all the data obtained by Shapley Value is subjected to the same regularization processing as ours.

To ensure the fairness of the measurement contribution in this experiment, we ensure that each agent is drawn the same number of times.

We randomly divided the data in the dataset, and distributed them evenly to all agents. Then, the corresponding special processing is performed on the data of the last few agents: reduce the amount of data by 30% and 70%, randomly generate the word sequence. Comparing the measurement results after special processing with the results when unprocessed data, it is concluded that the sensitivity of variables such as data size and data quality is evaluated.

### 4.5.1 EXPERIMENTAL RESULTS OF NORMAL MEASUREMENT

We did not do any special treatment to any agents, and randomly divided the data set into each agent. The result is shown in Fig. 2. In the figure, the abscissa is the index of agents, and the ordinate is the agents' contribution ratios. It can be seen that the deviation of each agent is not large, only individual agents are too high or too low. And the measurement result is similar to Shapley Value.

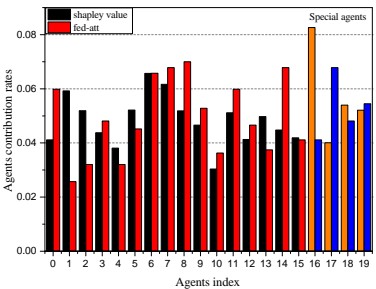
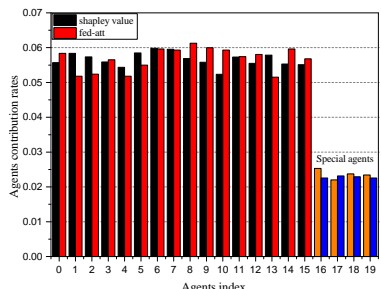

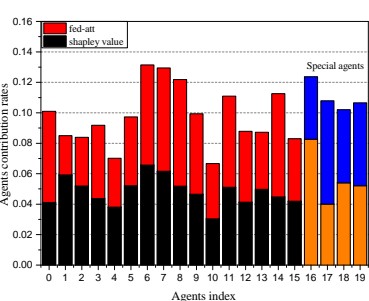
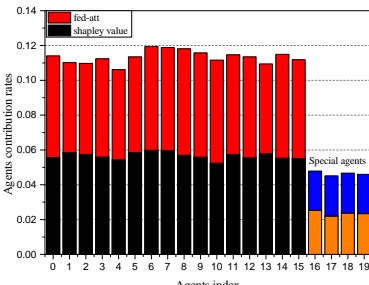

Figure 2: Experimental results of normal measurement.

Figure 3: Experimental results with randomize the word sequence.

### 4.5.2 EXPERIMENTAL RESULTS WITH RANDOMIZE THE WORD SEQUENCE

In this experiment, we modified the datasets of the next four agents into random word sequences. These should be regarded as dirty data by the model, so as to get a smaller return. The results are shown in Fig. 3. Both the method proposed in this paper and the Shpaley Value method can identify these bad agents and give them a small contribution.

### 4.5.3 EXPERIMENTAL RESULTS WITH REDUCE THE AMOUNT OF DATA

To demonstrate the sensitivity of our measurement method to the amount of data, we performed the following experiments.

In this experiment, we reduced the data amount of the last 4 agents by 70%, and the data amount of other agents remained unchanged. As you can see in Fig. 4, the contribution of the specially treated agents are significantly reduced. But it is relatively not obvious in the measurement results of Shapley Value.

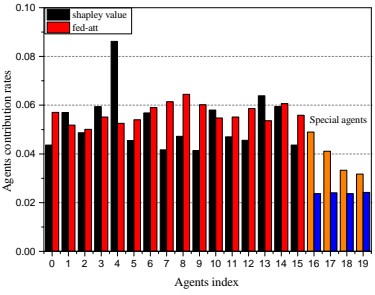 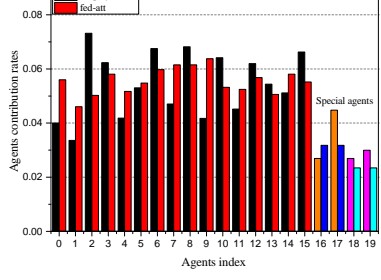

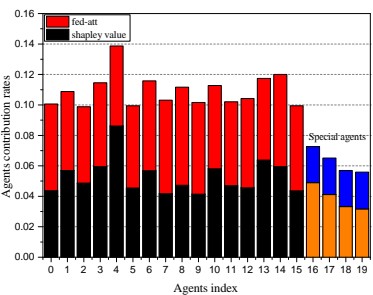 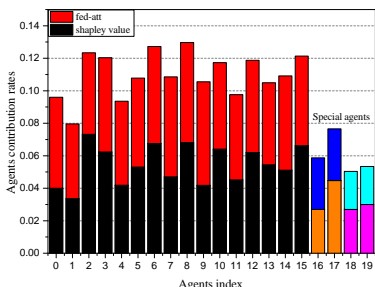

Figure 4: Experimental results with reduce the amount of data.
Figure 5: Experimental results with reduce the amount of data.

In order to reflect the relative relationship between the amount of data, we processed the data of the last 4 agents: agents with index 16 to 17 reduced the amount of data by 30%, agents with index 18 to 19 reduced the amount of data by 70%.

It can be seen in Fig. 5 that the method proposed in this paper can reflect the reduced amount of data, while Shapley Value cannot show it well. Note that the method proposed in this paper can better reflect the difference of smaller data volume.

## 5 CONCLUSION

A reasonable and fair completion of the measurement of the contribution of each participant in federated learning and the establishment of an incentive mechanism are essential for the development of federated learning. In this paper, we use a "FedAtt" method to train the models on Penn Treebank dataset. At the same time, we perform several special processing on the agents data, and compare the measurement results of the special processing with the measurement results of the unprocessed data. Experimental results show that we use this method to reasonably establish a measurement contribution mechanism to evaluate the sensitivity of indicators such as data size and data quality. The data is evaluated and trained by this method, and the calculated results are real-time and fast, and the contribution rate reflects accurately.

For future work, It is hoped that a better algorithm can be found that can more accurately measure the contribution to malicious agents, and at the same time prevent the contribution confusion caused by more than half of malicious agents. It also should consider that the negative contribution of agents measurement and the comprehensiveness of the measurement. Preliminary classification (i.e., positive and negative) is carried out at the end to avoid attacks under the federated learning framework mechanism(Bagdasaryan et al., 2018); at the same time, the concept of game theory is introduced, such as the establishment of a PVCG mechanism on the supply side(Cong et al., 2020). Improvements in these directions will promote the implementation and application of the federated learning incentive mechanism.

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
