# OpenReview forum: "A Real-time Contribution Measurement Method for Participants in Federated Learning"
_ICLR.cc/2021/Conference — Reject_

### Official Review · AnonReviewer4 · 2020-10-26
**Low computational complexity calculation of weights for federated learning clients**

**Rating:** 4
**Confidence:** 3

**Review:**

The paper proposes a low computational complexity method for weighting contributions of clients in a federated learning setting. The main contributions are to compare the weighting method with Shapley values and their sensitivity to low data volume and quality. The paper is based on the FedAtt paper that calculates weights based on the Euclidean distance between the server model and each client and for each layer.

The experimental setup is well described, including details about the hardware, software, datasets, model, and evaluation criteria. However, the model only specifies a "smaller GRU-based model" without giving any details of what that model is. They do not clearly describe some parameters of the approximation of the Shapley value calculation, reducing the value of the comparison between FedAtt and Shapley values. They could also have taken additional steps to improve the claims' confidence, e.g., only one dataset was used, which is relatively weak compared to the original FedAtt paper. The graphs in the results section could be described with more detail to explain what, e.g., the colors of the "special agents" mean. Also, there are no confidence measures specified, making it hard to evaluate the claims' validity.

The references include essential papers but are missing some core references, such as Federated Learning and Shapley values themselves. Also, related papers such as "Active Federated Learning" by Goetz et al. talk about very similar ideas but lack any mention in the paper.  The language and grammar could be improved, and some of the formulations make it hard to read. The comparison to Shapley values is also not motived in any detail, thus further reducing the paper contributions' value.

---

> ### Author Response · Authors · 2020-11-23
> **Reviewer 4 Response**
>
> Dear reviewer:
>
> Firstly, thank you for your review on our paper, we will accept it with an open mind and continue to improve the paper. We will strengthen the theoretical proof of the paper and provide more experimental data. The polishing of the paper is also one of the tasks we need to carry out, and the revision of the chart annotations. We will continue to enrich the experimental part of the problems on the experimental data set and refer to more recent research work. Thank you for your suggestions for our paper.

---

### Official Review · AnonReviewer3 · 2020-10-27
**The paper needs more thoughtful thinking.**

**Rating:** 3
**Confidence:** 5

**Review:**

The paper is to measure each client’s contribution to training the federated learning model. In particular, the contribution is measured by the distance between the local model and the global model in each iteration. The targeting problem is interesting, and the use of attention-based model divergence is also an interesting idea to measure the contribution. However, the paper lacks strict theoretical discussion to prove the proposed solution is a reasonable one rather than a heuristic method. Moreover, the experiment is too weak to support the claims. The paper’s technique contribution and originality are also limited.

Below are some detailed concerns.

1) The authors need to make a clear definition of the assumed application scenario so that the below problems can be avoided or solved.

If the client’s contribution is linked to rewards, it is unavoidable that some clients will produce fake data to gain more contribution to the commercial federation system. Therefore, the paper should discuss the prevention of “attacking by fake data”.

For example, if the client randomly shuffles the index of neurons in the trained local model w_k, then the client’s local model will get a bigger s_k^l calculated by equation 2. Thus, this client is likely to gain a big reward at every iteration.

According to equation 5, the contribution at the early stage will be discounted. It is unfair for the clients to be selected at an early stage. Therefore, from a systematic perspective, some clients may refuse to contribute to the training process at an early stage.


2) Contribution is not enough

The core method comes from the FedAtt algorithm – an attention-based federated aggregation method. The paper’s primary contribution relies on section 3.3 to measure the contribution according to the gradients.


3) The experiments are too weak to support their claim.

More datasets and baseline methods are required, for example, the FEMNIST, FeCeleba.

It is unclear how to define an objective metric to measure the quality of the proposed method. The contribution is a subjective feeling that various to different tasks and assessor.

---

> ### Author Response · Authors · 2020-11-23
> **Reviewer 3 Response**
>
> Dear reviewer:
>
> Thanks for your comments on our paper. We will improve the judgment and constraints of attackers, and strengthen the research on the system. At the same time, we will also select more datasets and baseline methods to enrich our experiments. Thanks again for your suggestions and review. We will continue to study related work.

---

### Official Review · AnonReviewer1 · 2020-10-28
**The idea lacks novelty and the experiments are not convincing**

**Rating:** 4
**Confidence:** 4

**Review:**

Summary:

The paper proposes a new contribution measurement approach for federated learning. The basic idea is that the agent with a larger model update has a larger contribution. Specifically, based on FedAtt [1], the impact of a client is computed as the local updates plus the impact of the previous round times a decay rate. The experiments on a dataset show that the proposed approach can have a similar contribution measurement compared with Shapley Value.

[1] Learning private neural language modeling with attentive aggregation.    IJCNN 2019

Strengths:

(1) The motivation of the paper is clear.

(2) The studied area is important. Effective incentive mechanisms in federated learning are still an open challenge.

Weakness:

(1) The proposed idea lacks novelty and may not be applicable in general federated learning algorithms. The contribution of each client is simply evaluated by its local update in FedAtt. FedAtt is not a widely used federated learning algorithm currently. It is not clear whether the proposed approach is applicable to other standard federated learning algorithms such as FedAvg. Also, I do not understand why the paper focuses on FedAtt instead of FedAvg.

(2) The paper lacks reasonable explanations for the proposed approach. A client may have arbitrary bad data and the local updated model may be far from the global optimal model. In such a case, since the distance between the local model and the global model is large, the contribution is also large according to the proposed approach, which is not reasonable. It is not clear how the proposed approach can handle such cases.

(3) The experiments are weak and not clear.

  a) It is not explained how the agent contribution rate is computed.

  b) The experiments are conducted on a single dataset. More datasets are needed.

  c) From Figure 2, it is hard to say that the proposed approach has a similar measurement with SV.

  d) Since the motivation is to reduce the computation overhead, the authors should show compare the computation complexity or the computation time of the proposed approach and SV.

Minor issues:

(1) The writing can be improved, e.g., “Such” -> “For example,”

(2) Figure 1 is not referred to in the text.

(3) Figure2-5: orange and blue colors are not explained.

---

> ### Author Response · Authors · 2020-11-23
> **Reviewer 1 Response**
>
> Dear reviewer:
>
> Thank you for your review on our paper, and thank you for your affirmation of the direction of our paper. We will choose more general algorithms for experimental verification, such as FedAVG and FedSGD. We will conduct theoretical proofs of our experiments based on game theory to enrich our theoretical part. For the problem of insufficient datasets, we will also verify with more datasets. In the meanwhile, we will continue to revise the issues of the paper to meet the requirements. Thank you again for your suggestions and affirmation of our work direction.

---

### Official Review · AnonReviewer2 · 2020-10-28
**A REAL-TIME CONTRIBUTION MEASUREMENT METHOD FOR PARTICIPANTS IN FEDERATED LEARNING**

**Rating:** 3
**Confidence:** 4

**Review:**

This paper designs an equation, i.e., equation (5) in the paper, to measure the impact or contribution of each participant/agent in federated learning. The designed measurement method is applied to attention aggregation algorithm of federated learning. Few experiments using Penn Treebank are conducted to support its claims.

This paper should be rejected because (1) the paper is unpolished and thus is hard to read, (2) the novelty appears quite weak, and (3) the experiments are difficult to understand and generally do not support its contributions

Concerns:

The paper is difficult to read due to the poor use of English. Many sentences are incomprehensible. Thus, it was often impossible for me to determine exactly what the authors would like to say or describe. Please have your submission proof-read for English writing style and grammar issues. Moreover, please treat the equations as the parts of sentences and make sure that the caption formats of Figures obey the ICLR format.

I also have a serious concern about the novelty of this paper. If my understanding is correct (due to the aforementioned reason), Subsection 3.3 is the only new material proposed by the authors. However, the proposed equation, i.e., equation (5), seems like a design choice without any theoretical justification or providing any intuitive reason, which significantly degrades the novelty of this paper.

Finally, the experiments should be refined to support its main claims. As claimed in Section 1, the proposed measurement method is real-time and has low computational complexity. However, no experiment nor quantitative comparison addressing the running time and complexity between the proposed method and Shapley Value. Actually, the authors compared their method with the method of approximating Shapley Value instead of exact Shapley Value. Furthermore, please cite for Shapley Value papers.

---

> ### Author Response · Authors · 2020-11-23
> **Reviewer 2 Response**
>
> Dear reviewer:
>
> Thank you for your review on our paper, we will accept it with an open mind and continue to improve the paper. In response to the dataset in the experiment, we will also verify it on more comprehensive dataset. Simultaneously, we will polish the sentence of the paper to make it easier to read. For the contribution measurement method proposed in this article, we will also follow strengthen the theoretical support. Based on our measurement method, we will compare it with the more accurate Shapley Value, which will be reflected in the subsequent submissions. Thanks again for your suggestions and review. We will continue to study related work.

---

### Decision · Program_Chairs · 2021-01-07
**Final Decision**

**Decision:**

Reject

**Comment:**

Although this paper tackles an important problem, all reviewers agree that it requires further work before it can be published. First, the paper would need to be polished in order to be easier to read. Stronger experiments would also be needed in order to support the claims of the paper, e.g. by considering additional datasets and proper baselines. Finally, an important concern about this paper is novelty and originality. It is not clear at this point that the contribution is substantial enough for a conference like ICLR. Addressing these points would significantly improve the paper.